# How to Choose? Comparing Different Methods to Count Wolf Packs in a Protected Area of the Northern Apennines

**DOI:** 10.3390/genes14040932

**Published:** 2023-04-18

**Authors:** Arianna Dissegna, Martino Rota, Simone Basile, Giuseppe Fusco, Marco Mencucci, Nadia Cappai, Marco Galaverni, Elena Fabbri, Edoardo Velli, Romolo Caniglia

**Affiliations:** 1Department of Biology, University of Padova, Via Ugo Bassi 58b, 35121 Padova, Italy; 2National Biodiversity Future Center (NBFC), Piazza Marina 61, 90133 Palermo, Italy; 3Reparto Carabinieri Parco Nazionale Foreste Casentinesi, Via G. Brocchi 7, 52015 Pratovecchio-Stia, Italy; 4Foreste Casentinesi National Park, Via G. Brocchi 7, 52015 Pratovecchio-Stia, Italy; 5Science Unit WWF Italia, Via Po 25c, 00198 Rome, Italy; 6Unit for Conservation Genetics (BIO-CGE), Italian Institute for Environmental Protection and Research (ISPRA), Via Cà Fornacetta 9, 40064 Ozzano dell’Emilia, Italy

**Keywords:** camera trapping, *Canis lupus*, conservation, non-invasive genetic sampling, protected areas, wolf howling, wolf packs

## Abstract

Despite a natural rewilding process that caused wolf populations in Europe to increase and expand in the last years, human–wolf conflicts still persist, threatening the long-term wolf presence in both anthropic and natural areas. Conservation management strategies should be carefully designed on updated population data and planned on a wide scale. Unfortunately, reliable ecological data are difficult and expensive to obtain and often hardly comparable through time or among different areas, especially because of different sampling designs. In order to assess the performance of different methods to estimate wolf (*Canis lupus* L.) abundance and distribution in southern Europe, we simultaneously applied three techniques: wolf howling, camera trapping and non-invasive genetic sampling in a protected area of the northern Apennines. We aimed at counting the minimum number of packs during a single wolf biological year and evaluating the pros and cons for each technique, comparing results obtained from different combinations of these three methods and testing how sampling effort may affect results. We found that packs’ identifications could be hardly comparable if methods were separately used with a low sampling effort: wolf howling identified nine, camera trapping 12 and non-invasive genetic sampling eight packs. However, increased sampling efforts produced more consistent and comparable results across all used methods, although results from different sampling designs should be carefully compared. The integration of the three techniques yielded the highest number of detected packs, 13, although with the highest effort and cost. A common standardised sampling strategy should be a priority approach to studying elusive large carnivores, such as the wolf, allowing for the comparison of key population parameters and developing shared and effective conservation management plans.

## 1. Introduction

During the last decades, most European wolf (*C. lupus*) populations have experienced a general numerical increase and geographical re-expansion, mainly due to favourable ecological conditions, adaptability of the species, protective legislation and a variety of human–wildlife coexistence practices [1]. However, some wolf populations are still locally threatened by anthropic causes, such as human persecution and anthropogenic hybridisation with domestic dogs [2,3,4], while new challenges are posed by particular situations, such as bold individuals habituated to human food sources. The impact of such threats on the long-term survival of current wolf populations should be carefully addressed to design adequate management and conservation measures based on reliable population data, such as abundance, distribution, pack number, home ranges, dispersal routes and hybridisation rates. Data collection should be primarily planned on a wide scale, especially when a target population extends its range across several administrative borders. However, such ecological data are difficult and expensive to obtain, and their comparison through time or among different areas is often impossible, mainly because of variations in sampling design and detection methods [5]. Indeed, the monitoring of wolf population dynamics can be achieved through a number of approaches, depending on aims, environmental conditions and available resources. In particular, methods can span from some of the earliest used, such as snow-tracking and wolf howling (WH), to relatively more recent ones, such as camera trapping (CT) and non-invasive genetic sampling (NGS) [6,7]. However, each of these methods has its advantages, specificity and limitations. Snow-tracking is seldom applicable in warm regions, and even in cold areas, the presence of snow is not always guaranteed or steady during the winter [8,9,10], particularly in recent years, due to global warming of the planet. WH could produce errors in group positions and false positives when used to detect packs with puppies, even when carried out by expert operators [11,12]. CT methods have some difficulties in individual identification and hybrid detection due to the usually low trapping success or lack of noticeable or recognisable physical characteristics [13]. NGS needs well-planned and protracted field sampling campaigns, might be affected by low genotyping success and high error rates, and could fail in intercepting all pack members [14]. Moreover, merging results from these different techniques can be challenging. Several ecological studies on wild mammals attempted to standardise sampling protocols and strategies, integrating different techniques and comparing their results and costs. Some of them showed that different applied methods could lead to similar outcomes, suggesting that the sampling protocol choice should be based on the available resources and ecological features of the study area [15,16]. Conversely, other studies found that only the combination of different methods could ensure reliable results [17,18,19,20]. Several research projects have been carried out so far to investigate the population ecology of the Italian wolf (*C. l. italicus*) and possible threats to its conservation status, mainly in the central-northern Apennines, through WH [21,22], CT [13,23] and NGS [24,25]. However, only a few pilot studies compared different concurrent methodological approaches [13,26].

In this study, carried out in a National Park of the northern Apennines, we simultaneously applied three commonly-used techniques to detect wolf presence (WH, CT and NGS) and compared the obtained results. In particular, we aimed at achieving a minimum number of wolf reproductive units (i.e., packs), which are good proxies, yearly updatable, of the basic units of demographic patterns [27,28]. Additionally, we evaluated the pros and cons for each of the tested techniques, their cost-effectiveness and how different sampling efforts might affect results.

## 2. Materials and Methods

### 2.1. Study Area

The study was performed in the Foreste Casentinesi National Park (FCNP, 43°480 N, 11°490 E), a 362 km^2^ protected area in Italy, located in the northern Apennines, between Tuscany and Emilia-Romagna regions, with altitudes ranging from 400 to 1658 m a.s.l. The area is featured by several water streams and one artificial lake, the Ridracoli Lake (3 km^2^). Snowfalls are occasional from November to April. Forests cover exceeds 80%, and the vegetation includes typical temperate-sub-Mediterranean species, such as beech (*Fagus sylvatica*) and Turkey oak (*Quercus cerris*). Low elevations are characterised by pastures and livestock breeding, whereas slopes and valleys are cultivated. The area is densely populated by wild ungulates: wild boar (*Sus scrofa*), roe deer (*Capreolus capreolus*), red deer (*Cervus elaphus*) and fallow deer (*Dama dama*). Human density is low, with about 2000 residents within the Park borders, showing a small numerical increase only during summer due to tourism (www.parcoforestecasentinesi.it, accessed on 15 December 2021).

### 2.2. Survey Design

Starting from the results of the last non-invasive genetic study performed in the monitored area from 2002 to 2009, which identified 8 wolf social units [24], in order to detect the current number of packs and to collect data in a uniform, standardised way, we applied a sampling grid with cells of 5 × 5 km to the whole study area, as suggested by [14]. We used the ETRS 1989 LAEA coordinate system, which is the base system suggested by the European Environmental Agency (EEA, 2006). We obtained a grid of 30 cells covering the entire territory of the FCNP (Figure 1). Among them, we selected an Increased Effort Sampling Area (IESA): nine contiguous cells selected for their good road connection and accessibility. In IESA, for each technique, we enhanced or doubled the sampling effort, as will be explained in detail later.

A sampling survey was performed from May 2019 to April 2020, following a wolf biological year, as suggested by [29]. Data collection involved more than 60 trained collaborators, including rangers of FCNP, researchers, students and volunteers.

### 2.3. Wolf Howling

We applied the WH technique (modified from [30]) from 22 July to 10 October 2019 (Table 1). We aimed to identify wolf packs through puppy replies, detecting rendez-vous sites (RVs), which are sites used by packs after the denning period [31]. We performed a total of 529 emissions in 44-night sessions (four nights a week and 12 emissions per night, on average) from 197 emission stations distributed in all of the 30 cells (6.6 stations per cell, on average). We chose the locations where wolves had been opportunistically detected by rangers of FCNP in previous years (Figure 2). We used three single-howl recorded emissions from a megaphone with a listening pause of three minutes between each emission and the last ten minutes of listening in order to detect puppies and estimate howl direction and distance. In addition to the data from systematic emissions, we also considered opportunistic detections of spontaneous howls. 

IESA WH sessions were performed on 29 evenings from 22 July to 10 October 2019 (Table 1), emitting 168 howls for each emission station (n = 85). In this area, we added one or more different locations with rangers as listening stations to improve detection reliability. Moreover, inside IESA, we placed seven camera traps set to take 60-s videos nearby the putative RVs found by WH to estimate the maximum distance covered by the cubs of the same pack. Camera-traps locations were chosen based on wolf faeces abundance [9]. We considered belonging to the same pack only the puppies’ howls detected within the maximum distance estimated by camera traps.

### 2.4. Camera Trapping

We placed 39 camera traps (1920 × 1080p Full HD video resolution, invisible LED, both AA batteries and 6V external batteries) near marking points along the same transects also used for non-invasive genetic sampling. We used one camera trap per cell, with an additional camera placed in each IESA cell (Figure 2). Camera traps were set to take 60-s videos and worked from 20 November 2019 to 10 April 2020 (Table 1) for a total of 4786 camera days. Each camera trap was inspected every 14 days and moved to another position along the transect in case it was not producing results.

Wolf videos were analysed blindly by three independently trained operators. We considered packs only groups of three or more individuals. Pack identifications were based on pack composition, camera traps positions and especially on peculiar individuals, called “focal individuals”, i.e., characterised by recognisable traits attributable to three main categories: (1) permanent characteristics (e.g., sex, tail or ear deformations, coat peculiarities); (2) temporary characteristics (e.g., scars, lameness, body size); (3) marking behaviour [13,23,32,33,34,35,36]. We considered as reliable only pack identifications with a 100% agreement among the three operators. For each identified pack, we considered as minimum pack size the maximum number of individuals captured in a single video [27,37].

### 2.5. Non-Invasive Genetic Sampling

We performed NGS following two different approaches: from 1 May 2019 to 30 November 2019, we collected faecal samples only opportunistically; from 1 December 2019 to 31 March 2020, we collected faecal samples through a systematic sampling design (Table 1) [38]. During systematic sampling, we walked 39 transects (each one of about 5 km), at least one per cell, twice per month for four months, for a total effort of about 1600 km. In IESA, we set two transects per cell (each one of about 5 km; Figure 2). Faecal samples were collected using a faecal-swab protocol [39], focusing on hydrated samples possibly fresher than two weeks [26]. We stored samples at 5 °C inside 2.0 mL safe-lock tubes containing 300 µL of ATL lysis buffer (Qiagen Inc., Hilden, Germany). In the study area, during the period May 2019/April 2020, we also opportunistically sampled additional biological samples, such as muscle tissues from wolves found dead or blood samples from rescued individuals.

Total DNA was extracted from collected faecal samples following the procedures reported in [24]. Each DNA sample was amplified at 12 unlinked autosomal canine microsatellite loci (short tandem repeats—STR), which allow to reliably identify individual genotypes (PID = 8.2 × 10^−6^; PID_sibs_ = 7.3 × 10^−3^) and which well-differentiate wolves, dogs and their first two generation hybrids in assignment procedures [24,40,41,42]. Paternal haplotypes were identified in males using four Y-chromosome microsatellites (MS34A, MS34B, MSY41A and MS41B) [43] to confirm the taxon identification or, in the case of admixed individuals, to provide the directionality of hybridisation, while coat colour patterns were inferred through the sequencing of a dominant 3-bp deletion at the β-defensin CBD103 gene (the K-locus) associated with black coat colour [44,45]. The software GIMLET 1.3.3 [46] was used to reconstruct the consensus genotype from the results of the 4–8 replicated amplifications per locus, to estimate the genotyping error rate (allelic drop-out and false allele) and to match the detected genotypes to each other and to the Italian Institute for Environmental Protection and Research (ISPRA) *Canis* database for the identification of possible re-samplings. Finally, a Bayesian clustering procedure on the 12-loci multi-locus reliable genotypes was performed using PARALLEL STRUCTURE software to identify individuals as wolves, dogs, or admixed based on their individual proportions of membership [25,41,47,48]. Further details on the amplification protocols and statistical procedures for the genotype assignment to the wolf, dog or admixed populations are reported in (S1) [24,42].

We mapped spatial distributions of the genetically identified individuals using minimal bounding geometries in QGIS 3.2.3. We considered as hypothetical packs distinct groups of individuals and/or single individuals sampled at least two times within a maximum area of 100 km^2^, considered as signs of the possible presence of marking individuals [24]. For each hypothetical pack, we traced a 100 km^2^ circular buffer, including the sampling locations of their genotypes. We reconstructed the genealogy of each hypothetical pack using a maximum-likelihood approach implemented in the software COLONY 2.0 [49]. For each hypothetical pack, we considered as candidate parents all the males and females, excluding those individuals who died before reproductive maturity (when this information was available); as candidate offspring, all the individuals. We ran COLONY with the ‘monogamous’ breeding system, allele frequencies and PCR error rates estimated from all the analysed genotypes with GENALEX 6.1 [50] and GIMLET 1.3.3 [46], and assuming a 1.0 probability of including fathers and mothers in the candidate parental pairs, excluding father-son combinations with incongruities at Y-STR haplotypes. To further test for all possible parentage combinations, we compared the best maximum-likelihood genealogies to those obtained by an ‘open parentage analysis’ also performed in COLONY, using all the individuals collected as one group. We selected output pedigrees from COLONY showing a pair of genotypes with a probability *p* > 0.90 to be parents of one or more offspring within the 100 km^2^ hypothetical pack’s area. We identified as putative dispersers individuals belonging to a family group but then sampled outside the 100-km^2^ hypothetical pack’s area [24].

### 2.6. Comparing and Integrating Methods 

After founding the number and approximate location of packs with each sampling technique, we also tested the effectiveness of four possible techniques combinations to verify if integrated methods would identify more packs: (1) WH + CT; (2) WH + NGS; (3) CT + NGS; (4) WH + CT + NGS. For all packs identified with each technique, we drew a minimum convex polygon when three or more localisations were available. We then considered overlapping pack identifications from two or three different techniques as the same pack and rated them as A (most reliable). In order to avoid overestimation, if no spatial overlapping among pack identifications occurred, we merged packs identified with any technique in an area smaller than 100 km^2^. These identifications were rated as B. Other kinds of pack identifications were rated as C, the less reliable.

### 2.7. Costs and Working Time

We calculated the costs of the different methods used in this study, considering the necessary equipment for data collection and analysis. For WH, the equipment consisted of megaphones, SD cards with recorded howls and compasses. CT required camera trap devices, including Python™ cables, SD cards, batteries and padlocks. NGS included all genotyping costs (reagents, plastics and machine usage). We did not consider transportation costs and salaries of personnel, as they are highly variable and dependent on who was involved and in which part of the study area. We counted instead man-power time needed for data collection and analysis, rounded to full days.

## 3. Results

### 3.1. Wolf Howling

We obtained a total of 60 wolf replies (corresponding to 11.3% of emissions) from 41 stations in the study area (20% of the total number) in 17 cells (56.7%). Among wolf replies, 39 (65.0%) included puppy howling from 24 stations (58.5%) in 11 cells (36.7%, Table 2). We added to these results eight litter localisations from spontaneous howls detected during summer and the FCNP deer census activities at the end of September. 

The supplementary CT survey identified four litters, with four puppies each. The maximum distance between different locations of the same litter was 3.41 km (Table 3, Figure 3). Using this distance as a threshold value, we identified a minimum number of nine packs (Figure 4a). The average distance among the same litter localisations found through WH was 2.32 km (min = 1.68, max = 3.36, Table 3). 

Considering only IESA, we obtained 33 replies (19.6% of emissions) from 18 stations (21.2%) in eight cells (88.8%), and among these answers, 21 (63.6%) included puppies, heard from 13 different stations (72.2%) in five cells (55.5%). In IESA, we identified four packs (44.5% of the total number) in five cells (55.6%, Table 2).

### 3.2. Camera Trapping 

We obtained 1464 videos of wolves from 39 camera traps in the study area in 4786 camera days (37.5 wolf videos/camera trap; a trapping rate of 0.306 videos per camera day). Five cameras (12.8%) were stolen but replaced within one month. In 28 out of 30 cells, we documented at least once a group of three wolves (Table 2). We filmed, on average, 2.2 wolves per video (min = 1, max = 12) and recorded groups of three or more wolves in 389 videos (26.6%). In 539 wolf videos (36.8%), there were individuals with some useful characteristics for individual or pack identification: 395 (73.3%) of them led to the identification of a pack, while we recognised two possible couples and nine dispersers in the other 144 videos. Using an average of 33.1 videos per pack and 24 focal individuals (Table 4), all operators identified 12 packs, with 100% of agreement (Figure 4b) in 22 cells (73.3%, Table 2). Four packs had focal individuals with permanent characteristics, such as a black coat or tail deformation (Figure 5). Six packs had focal individuals with temporary characteristics, such as mange or lameness, or particular marking behaviours, like young individuals always participating in marking activities. Only two packs did not present focal individuals, and they were identified due to differences in pack composition compared to other neighbouring recognisable packs. The average pack size was 7.2 individuals (min = 3, max = 12; Table 4). 

Considering only IESA, we recorded 767 wolf videos (52.4% of the total number). Of these, 214 videos (27.9%) documented groups of three or more wolves. Videos with focal individuals were 281 (36.6%), and those that led to a pack identification were 199 (70.8%), thanks to 7 focal individuals. In IESA, we found five packs (41.7%) in nine cells (100%, Table 2). 

### 3.3. Non-Invasive Genetic Sampling 

We collected a total of 294 faecal swab samples across the study area, 9.8 samples per cell on average (min = 1, max = 24). We successfully genotyped 148 samples (50.3%), identifying 93 unique genotypes belonging to two dogs and 91 wolves (39 males, 50 females and 2 undetermined). In particular, two male wolves were classified as recent wolf-dog hybrids (within second-generation backcross). Another male showed the 3-bp melanistic deletion at the K-locus [45], and five other males showed dog Y-haplotypes, suggesting a past introgression of domestic allelic variants along the paternal lineage. We detected, on average, each individual 1.6 times: 66 individuals only once (72.5%) and 25 from 2 to 12 times (27.5%). To this dataset, we added another eight genotypes (three males and five females) obtained from tissue samples collected from found dead wolves. Through individuals’ spatial distribution, we found 12 hypothetical packs. Parentage analyses allowed us to confirm the presence of at least eight packs (Figure 4c) in 20 cells (66.7%, Table 2). The average offspring number was 3.8 (Table 5). Among tissue samples from found dead wolves, there was also a female (W2373F) found in cell 28, pregnant with four foetuses, which resulted in being a daughter of pack H. In two packs (B and F), both reproductive females probably dispersed from pack C. 

Considering only IESA, we collected 141 samples (47% of the total number), 15.6 samples/cell on average (min = 9, max = 24). We obtained 42 wolf genotypes (46.1% of the total number): 19 males, 22 females, and one undetermined. We detected 13 individuals two times or more (52% of the total number). In IESA, we found five packs (62.5%) in nine cells (100%, Table 2).

### 3.4. Integrated Methods 

The integration of WH/CT/NGS allowed us to overall identify 13 packs (Figure 6a) in 28 cells (93.3%, Table 2), with six packs rated as A (46.2%), three as B (23.1%) and four as C (30.7%, Table 6). Considering WH/NGS, we found ten packs (Figure 6b) identified in 22 cells (73.3%, Table 2), with three rated as A (30.0%), four as B (40.0%) and three as C (30.0%, Table 6). Through both WH/CT and CT/NGS combinations, we identified 12 packs (Figure 6c,d). The WH/CT combination found packs in 25 (83.3%) cells, whereas CT/NGS in 24 (80.0%, Table 2). WH/CT identified four packs rated as A (33.3%), five as B (41.7%) and three as C (25.0%), while CT/NGS combination found six packs as A (50.0%), two as B (16.7%) and four as C (33.3%, Table 6). 

Considering only IESA, all different combinations led to the identification of the same five packs in all nine cells (Table 2). The rating was mostly A: with WH/CT/NGS and CT/NGS, we rated all five packs with the highest score; the combination of WH/CT led to four packs rated as A (80.0%), and only one rated as B (20.0%); WH/NGS had three packs rated as A (60.0%) and two rated as B (40.0%, Table 6).

### 3.5. Costs and Working Time 

Costs were different comparing the three techniques (Appendix A). WH was the cheapest method, requiring only 720.96 €, while CT and NGS were about 17 and 41 times more expensive than WH, requiring 12,168.80 € and 29,400 €, respectively. Considering manpower time, too, WH was confirmed to be the technique with the lowest effort (Table 2), with only 72 workdays. NGS was the most demanding technique, with 506 workdays in about 7 months (4 months for sampling and 3 for data analysis). CT required 412 workdays in 6 months: the same 4 sampling months as for NGS, with 2 months for data analysis. 

## 4. Discussion

The knowledge of key population parameters (i.e., abundance or density estimates, the minimum number of resident individuals, dispersers, and floaters) is a primary goal in designing effective management and sound conservation strategies for threatened taxa [51]. However, sampling strategies and analysis protocols should be objective, reliable, economically affordable and easily reproducible over time and in different areas. For large elusive carnivores, such as the wolf, obtaining reliable population data might be far from trivial, often requiring the integration of different techniques, which is demanding in terms of resources and time. However, such an integrated approach can be affordable at local scales, dealing with sub-populations or in limited geographical areas [9]. Thus, as suggested by [27], researchers studying pivotal aspects of wolf population ecology can usefully focus on a “reproductive unit scale” in order to collect data not only about single wolves but also on pack presence, distribution and structure in local contexts that could be then extended to the entire “population range scale”. Our study followed this approach, using a reproducible sampling design, with the aim of providing reliable and applicable guidelines for future wolf ecological and population dynamic monitoring programs. We carried out a survey using three common techniques for monitoring wolf ecology and pack dynamics in the Apennines (WH, CT, NGS), in order to count the minimum number of social units detected during a single wolf biological year in a protected area of the northern Apennines. 

Our reply rates during WH surveys (11.3%) were close to those obtained in similar studies (12.6% in [52]; 14% in [21]). More than half of the replies in the whole study area (65%) included puppies. This is in line with previous studies confirming late summer as the best period for detecting wolves through this technique due to puppy presence and their greater tendency to reply [30,52,53]. We found a maximum reference distance of 3.4 km between two RV locations used by one pack. This value is very close to the maximum reference distance of 3.8 km suggested by [30]. With WH, we identified nine packs, detected in 40% of the cells, as expected, considering that this technique was used only during summer. In this period, wolf territories tend to shrink around RVs, mainly to feed and protect their offspring [31,54]. This implies that wolves in summer are more likely to be detected only close to their RVs [9,30]. In IESA, our reply rates (19.6%) and study area coverage (55.6%) were higher compared to the whole study area, likely due to the sampling approach in IESA, with one or more listening teams per emission station, which could have improved the detection capability, since the human range of hearing howls is approximatively 3.2 km [30] while wolf’s hearing may span from 10 to 16 km [31]. The number of detected packs with WH (9) was close to NGS (8) and lower than CT (12 packs). The lower detection of packs compared to CT could be due to false negatives. In fact, wolves do not necessarily always reply to emissions, even if they are present [30]. Moreover, considering how this technique tries to detect packs’ reproduction and puppies’ replies, we have to consider that packs may also fail to reproduce for one or more years. Indeed, puppy mortality is very high during the first months of life [31]. In IESA, we detected with WH the lowest number of packs (4) compared to all other techniques (5). However, by overlapping different results, we can see that the fifth pack “missed” by WH had, in fact, the RV outside IESA. This means WH actually identified the same packs as the other techniques but partially outside IESA.

The frequency of wolf videos in our study was 0.306 videos per camera day, much higher than in other similar wolf CT studies performed in the Apennines (0.08 [26]; 0.22 [13] and 0.17 [23]), despite the several thefts of camera traps, a widespread problem for research groups, which could have limited the overall number of successful wolf recordings [55]. In our study, 73.3% of wolf videos led to possible pack identifications thanks to the detection of 24 focal individuals. These values about focal individuals are comparable to those obtained in similar canid studies: 76.8% for the red fox in Spain [36]; 81% for the Apennine wolf [23]. We identified 12 packs in 73.3% of the cells, with an average minimum pack size of 7.2 wolves/pack, for a total minimum number of 86 wolves. These are larger numbers considering the only comparable study carried out close to our study area in Arezzo province, with 3.4 wolves/pack in 2014 and 4.2 wolves/pack in 2015, for a total of 43 and 50 wolves [23]. However, that study used CT mainly in spring, when the pack size can be smaller due to the dispersion of young individuals during late winter [14]. This highlights the importance of accounting for sampling periods when designing a CT survey. In IESA, videos led to the identification of five packs in 100% of the nine cells. This percentage could result from the use of two camera traps per cell instead of one, which doubled in every IESA cell the detection probability. CT counted the highest number of packs among the applied techniques. However, three packs (P04, P08 and P10) were detected only by one camera site each at the FCNP borders (P04 in cell n°8; P08 in cell n°18; P10 in cell n°23). This is concordant with [23], who found 12 packs close to FCNP, in Arezzo province in Tuscany, with three of them detected only by one camera site each at the border of their study area. Considering that both studies focused on marking points, CT could have also detected some pack territories neighbouring the study area. In fact, even with integrated methods, two packs (P04 and P08) were always rated as C and detected only by CT.

In our study, NGS genotyping success rate was 50.3%, close to what was found in other wolf NGS studies (52.8% [56]; 45% [57]; 48.5% [26]; 41% [58]). Differently from similar studies [24,26,56], we detected a slightly higher number of females (56.2%) than males (43.8%). We identified eight packs in 66.7% of the study area cells. The average offspring number per pack was 3.8, combined with the parental couple and excluding disperser individuals, which provides an average minimum number of 5.3 wolves per pack, which is very close to the findings (5.6) from a previous study carried out in the same sector of the Apennines, including our study area [24]. Interestingly, the total number of 99 detected genotypes is very close to the CT total number of 97 individuals (86 from identified packs, plus two couples and nine dispersers recognised). In Idaho [59], researchers have already found comparable NGS and CT results in terms of estimations of wolves’ numbers. On the other side, with CT, we obtained a higher average number of individuals per pack than NGS (7.2 vs. 5.3 wolves/pack). This could be due to the unrelated adoptees not being identified within the pack genealogy [24] or to the different marking behaviour in young and subordinate individuals, which usually do not intensively mark the territory, leading to more likely missing some of them during sampling activities [56].

Overall, NGS alone led to the identification of the smallest number of packs (8) among single techniques, although IESA was able to detect the same number of packs of other methods and their combinations. This reminds the importance of well-evaluating the minimum sampling effort for NGS in order to provide robust results. Furthermore, we should take into account that NGS performs best, especially on long-term monitoring programs, where the chances to sample a whole family group increases with sampling sessions [14,24,56,60]. However, NGS is the only technique that can provide valuable information that could not be obtained in other ways, such as hybridisation and detailed pack dynamics. In fact, only through NGS we detected that some offspring found or reproduced in different packs from their natal one, showing female wolves’ tendency to stay closer than males to their natal pack territories, as documented in many studies [24,31,60,61,62]. Moreover, although wolf *x* dog hybridisation represents a widespread phenomenon in many areas of the Apennines [13,23,63], in our survey year, only two individuals (2.0%) were identified as recent hybrids, similar to the situation described in Croatia (2.8% [2]). CT detected only one of the putative admixed individuals confirmed by NGS (a wolf with a black coat), as hybrids do not always show anomalous diagnostic phenotypes [64,65]. These findings confirm the importance of using both NGS and CT techniques simultaneously when dealing with deeply introgressed wolf populations [13,23]. 

The combinations of techniques WH/CT and CT/NGS found the same numbers of packs as using CT alone (12), while WH/NGS identified fewer packs (10). The three techniques combination (WH/CT/NGS) showed almost a total coverage of the study area (93%) and the highest number of packs detected (13), among which six were rated as A due to a good spatial overlapping of the identified territories across methods. In IESA, the three integrated methods identified the same five packs as the single techniques applied alone in 100% of the nine cells. This indicates that a greater sampling effort for each technique is fundamental to get closer to the actual number of packs, and with sufficient sampling, different techniques are able to provide comparable results [59].

From the economic point of view, WH was the least expensive technique, while CT costs were less than half the costs necessary for NGS. Conversely, despite being the most resource-demanding, NGS yielded the lowest number of identified packs. However, consistently with other studies, NGS provided much other exclusive information: minimum number of individual genotypes [57]; gene flow among different packs [62]; multiyear and multipack genealogies [24,61]; wolf-dog admixture evidence [45,65,66,67,68]. As expected, the integrated approaches were more expensive but identified a major number of packs providing more consistent and comparable results, as well as offsetting drawbacks from single methods alone. 

However, our results were obtained from a single wolf biological year in a single study area. Therefore, these results should be implemented in additional future studies, applying this survey design for several reproductive years and replicating it in different areas. Furthermore, other future perspectives in wolf pack detection should be considered, such as the support of telemetry [69], drones [70] or improved molecular tools, such as specific panels of single nucleotide polymorphisms, highly performing in genotyping non-invasively collected samples (SNPs) [71,72] and innovative environmental DNA techniques [73].

Nonetheless, in our pilot study, we managed to show how wolf pack identifications can be hardly comparable if different survey methods are singularly used with a low effort. Such an issue significantly decreases when two or more techniques are combined and almost disappears with a higher sampling effort, yielding consistent and comparable results, even with more affordable costs. Therefore, an optimal combination of techniques and sampling efforts should be evaluated before starting each study, taking into account available human and economic resources, environmental features, sampling interval and period, as these variables can strongly affect the obtained results and their usefulness for conservation and management purposes.

## Figures and Tables

**Figure 1 genes-14-00932-f001:**
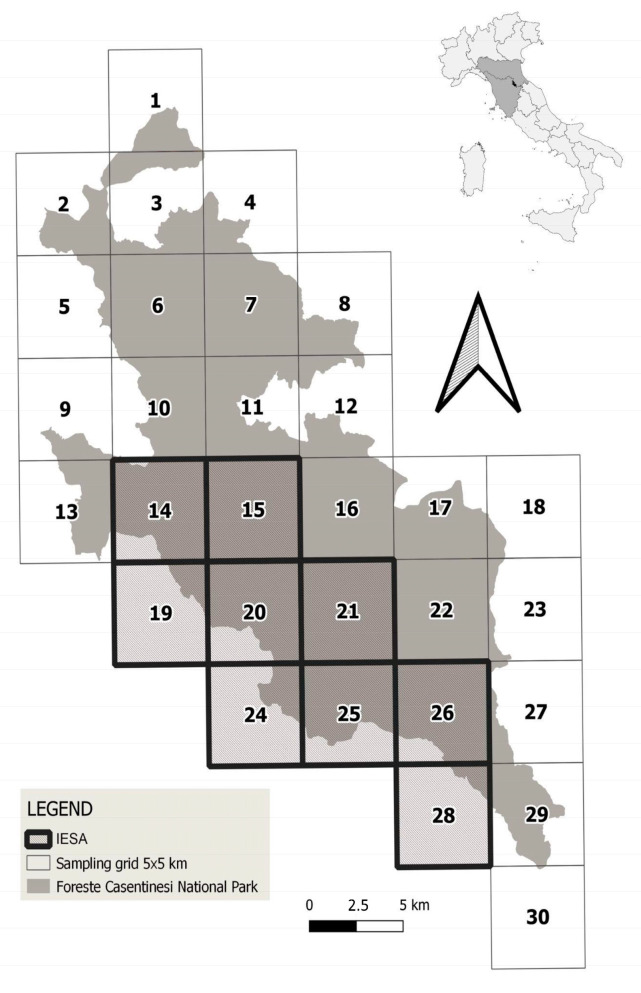
Sampling grid 5 × 5 km across the Foreste Casentinesi National Park in the Northern Apennines, Italy. Cells are numbered from 1 to 30. Increased Effort Sampling Area (IESA) cells are indicated in bold.

**Figure 2 genes-14-00932-f002:**
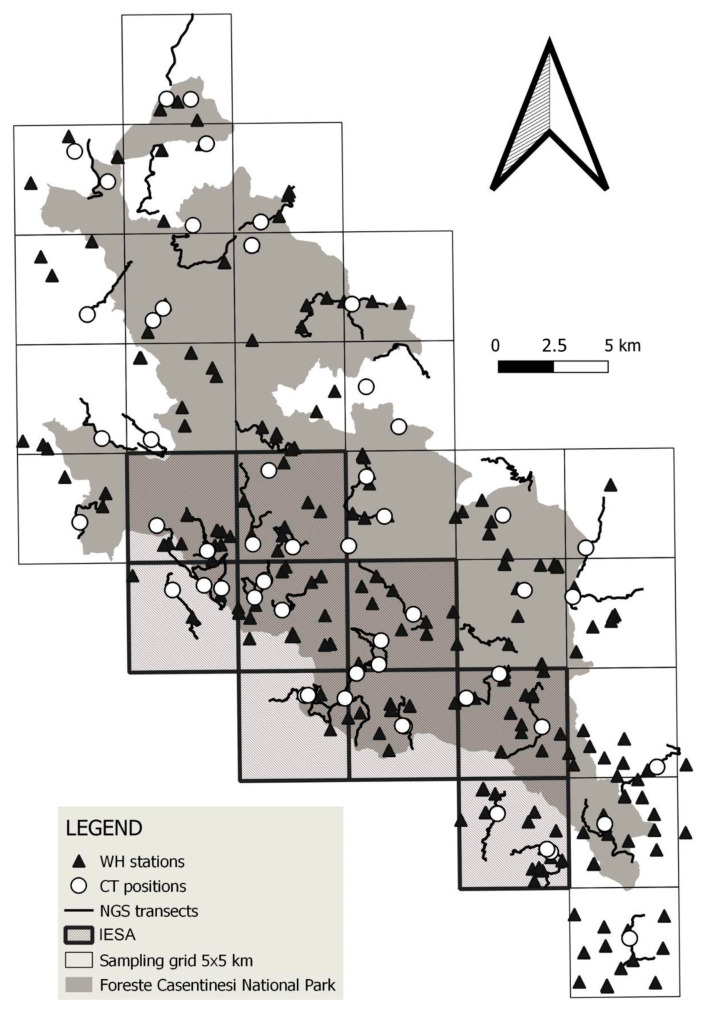
Sampling effort in the study area. The legend shows the types of sampling sites: wolf howling (WH), camera trapping (CT), and non-invasive genetic sampling (NGS).

**Figure 3 genes-14-00932-f003:**
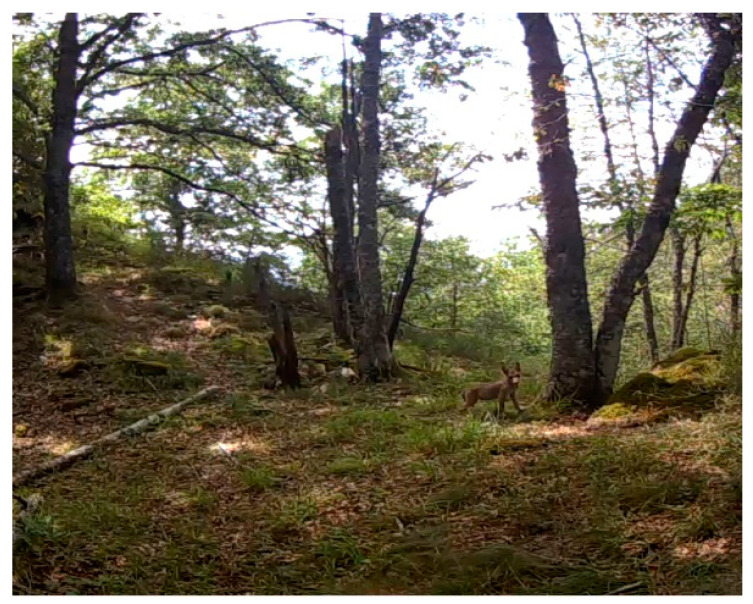
Rendez-vous site of TR pack. This picture was taken when this pack was still localised only on one site. In a few weeks, the reproductive female (Figure 5c) would have been seen with puppies in another area, 3.4 km from this place. For a few days, the litter was seen simultaneously split in two places.

**Figure 4 genes-14-00932-f004:**
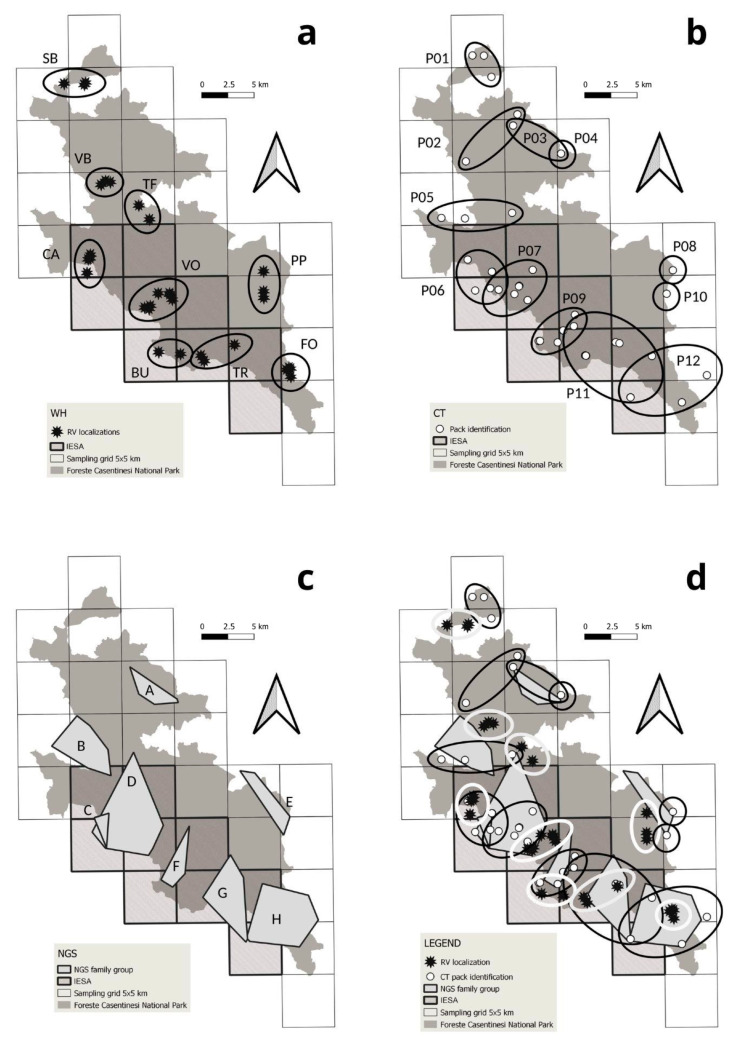
Wolf packs’ identifications with different techniques: (**a**) with wolf howling, we identified 9 packs (named after locality abbreviations); (**b**) with camera trapping, we identified 12 packs (named from P01 to P12); (**c**) with non-invasive genetic sampling, we identified 8 packs (named from A to H); (**d**) spatial comparison among different results.

**Figure 5 genes-14-00932-f005:**
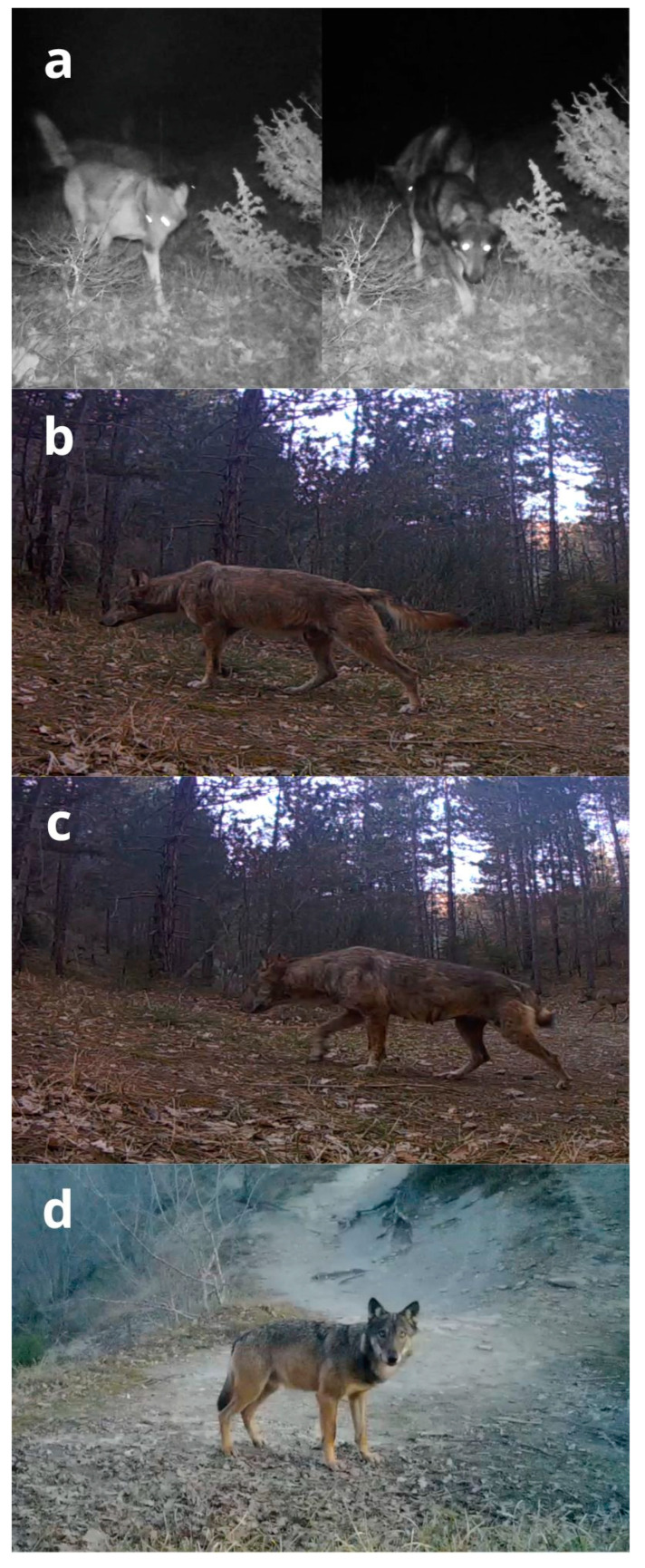
Screenshots from videos made during CT systematic sampling: (**a**) The dominant couple of P08, a wildtype female (on the left) and an almost completely black male (on the right), probably due to past wolf x dog hybridisation; (**b**,**c**) The dominant male (**b**) and the dominant female (**c**) of pack P11, both with mange during winter, a temporary characteristic; it is also possible to see in the picture of the dominant female one of her permanent characteristics, the short tail with a deformation on the top. (**d**) A healthy male wolf with a typical wild-type phenotype, for comparison.

**Figure 6 genes-14-00932-f006:**
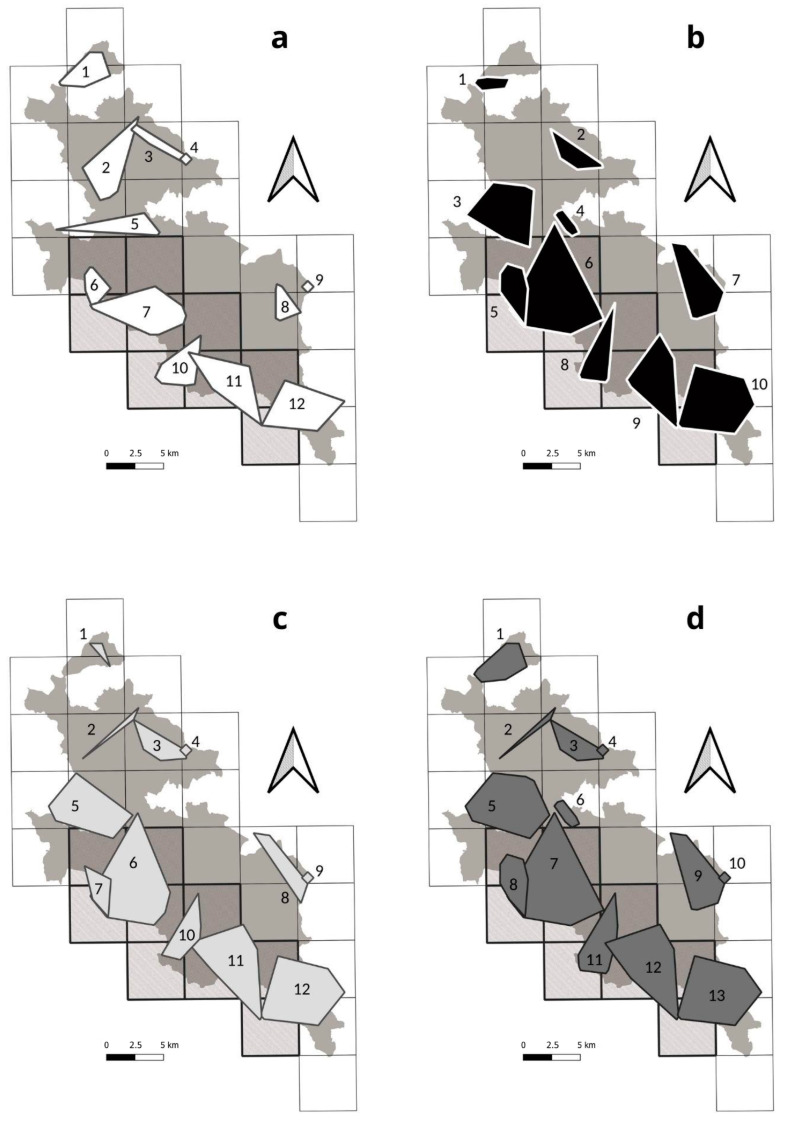
Wolf packs identified by integrated methods and numbered from 1 to 13: (**a**) wolf howling and camera trapping; (**b**) wolf howling and non-invasive genetic sampling; (**c**) camera trapping and non-invasive genetic sampling; (**d**) wolf howling, camera trapping and non-invasive genetic sampling.

**Table 1 genes-14-00932-t001:** Sampling activities from May 2019 to April 2020: WH = wolf howling, CT = camera trapping, NGS = non-invasive genetic sampling.

2019								2020			
May	Jun	Jul	Aug	Sep	Oct	Nov	Dec	Jan	Feb	Mar	Apr
		WH	WH	WH	WH						
		CT	CT	CT	CT	CT	CT *	CT *	CT *	CT *	CT
NGS	NGS	NGS	NGS	NGS	NGS	NGS	NGS *	NGS *	NGS *	NGS *	NGS

* Systematic sampling design.

**Table 2 genes-14-00932-t002:** Yield (area coverage and packs) and effort (costs and workdays) of each technique: wolf howling, camera trapping, non-invasive genetic sampling and all integrated methods.

	Area Coverage				Packs				Effort	
Technique	% Cells with Sampling Results *		% Cells with Packs Identification		N° Packs		Packs Rated as A		Costs (€)	Total Work Days
WH	36.7	55.6	40	55.6	9	4	-	-	720.96	72
CT	93.3	100	73.3	100	12	5	-	-	12,168.80	412
NGS	100	100	66.7	100	8	5	-	-	29,400.00	506
WH + CT	93.3	100	83.3	100	12	5	4	4	12,889.76	484
WH + NGS	100	100	73.3	100	10	5	3	3	30,120.96	578
CT + NGS	100	100	80	100	12	5	6	5	41,568.80	606 **
WH + CT + NGS	100	100	93.3	100	13	5	6	5	42,289.76	678 **

* For WH: area with detection of puppies. * For CT: area with detection of a group of at least three wolves. * For NGS: area with genotyped individuals. ** CT and NGS sampling can be performed on the same work day if camera traps are along the transects. Increased Effort Sampling Area = IESA.

**Table 3 genes-14-00932-t003:** Information on pack litters identified by wolf howling and confirmed in IESA by camera trapping.

Wolf Howling (WH)				Camera Trapping (CT)			
Pack ID	N° Localizations (WH)	Distance (km)	Detection Days	N° Localizations (CT)	Distance (km)	Detection Days	N° Puppies (CT)
SB	2	2.06	24/07/2019 04/09/2019	-	-	-	-
VB	1	-	26/09/2019	-	-	-	-
TF	2	1.68	19/08/2019 09/09/2019	-	-	-	-
CA	2	1.78	01/08/2019 07/08/2019	2	1.94	From 25/09/2019 to 30/09/2019	4
PP	2	2.51	03/08/2019 27/08/2019 05/09/2019	-	-	-	-
VO	2	2.75	23/07/2019 19/08/2019	-	-	-	-
BU	2	2.09	29/07/2019 22/08/2019	1	-	From 29/08/2019 to 18/09/2019	4
TR	2	3.36	31/07/2019 15/08/2019 23/08/2019	2	3.41	From 02/08/2019 to 28/09/2019	4
FO	1	-	25/07/2019	1	-	From 19/08/2019 to 28/09/2019	4
Average	1.8	2.32	-	1.5	2.67	-	4

**Table 4 genes-14-00932-t004:** Information on packs identified by camera trapping: number of videos per pack, minimum pack size, the minimum number of females and males identified, the temporal range of detections, number of focal individuals and their characteristics.

Camera Trapping (CT)							
Pack ID	N° Videos	Pack Size	N° Females Detected	N° Males Detected	First Detection	Last Detection	N° Focal Individuals
P01	12	6	3	1	03/12/2019	20/02/2020	2
P02	34	8	1	1	25/11/2019	02/04/2020	2
P03	22	3	1	1	04/12/2019	19/03/2020	1
P04	31	8	2	1	01/12/2019	12/03/2020	3
P05	19	9	-	-	21/12/2019	05/03/2020	4
P06	9	9	1	1	12/12/2019	29/03/2020	2
P07	25	6	2	3	25/11/2019	02/04/2020	3
P08	18	6	2	1	21/11/219	08/02/2020	5
P09	70	7	2	1	07/12/2019	07/04/2020	1
P10	13	3	1	1	21/01/2020	30/03/2020	0
P11	76	12	3	2	03/12/2019	01/04/2020	1
P12	68	9	2	2	24/11/2019	15/03/2020	0
Average	33.1	7.2	1.8	1.4	-	-	2

**Table 5 genes-14-00932-t005:** Information on packs identified by non-invasive genetic sampling.

Non-Invasive Genetic Sampling (NGS)						
Pack ID	Pack Size	Reproductive Couple	N° Offspring	N° Female Offspring	N° Male Offspring	N° Dispersers
A	4	HFO3M + WFO313F	2	1	1	0
B	5	WFO229M + WFO239F	4	1	3	1
C	3	WFO204M + WFO315F	3	3	0	2
D	6	WFO288M + WFO249F	4	1	3	0
E	4	WFO278M + WFO282F	2	1	1	0
F	5	WFO251M + WFO205F	3	3	0	0
G	8	WFO232M + WFO206F	6	4	2	0
H	7	HFO1M + WFO256F	6	5	1	1
Average	5.3	-	3.8	2.4	1.4	0.5

**Table 6 genes-14-00932-t006:** Packs identified by integrated methods. Packs’ IDs with two letters are from WH; packs’ IDs with P plus a number are from CT; packs’ IDs with one letter are from NGS.

N° Packs	WH + CT	Grade	WH + NGS	Grade	CT + NGS	Grade	WH + CT + NGS	Grade
1	SB + P01	(B)	SB + --	(C)	P01 + --	(C)	SB + P01 + --	(B)
2	VB + P02	(B)	-- + A	(C)	P02 + --	(C)	-- + P02 + --	(C)
3	-- + P03	(C)	VB + B	(B)	P03 + A	(B)	-- + P03 + A	(B)
4	-- + P04	(C)	TF + --	(C)	P04 + --	(C)	-- + P04 + --	(C)
5	TF + P05	(B)	CA + C	(B)	P05 + B	(A)	VB + P05 + B	(A)
6	CA + P06	(A)	VO + D	(A)	P06 + C	(A)	TF + -- + --	(C)
7	VO + P07	(A)	PP + E	(B)	P07 + D	(A)	VO + P07 + D	(A)
8	PP + P10	(B)	BU + F	(B)	P10 + E	(B)	CA + P06 + C	(A)
9	-- + P08	(C)	TR + G	(A)	P08 + --	(C)	PP + P10 + E	(B)
10	BU + P09	(B)	FO + H	(A)	P09 + F	(A)	-- + P08 + --	(C)
11	TR + P11	(A)	-- + --	--	P11 + G	(A)	BU + P09 + F	(A)
12	FO + P12	(A)	-- + --	--	P12 + H	(A)	TR + P11 + G	(A)
13	-- + --	--	-- + --	--	-- + --	--	FO + P12 + H	(A)

## Data Availability

The data presented in this study are available on request.

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
