# Peer review of "How to Choose? Comparing Different Methods to Count Wolf Packs in a Protected Area of the Northern Apennines"

_genes, 2023, doi:10.3390/genes14040932_

Round 1
Reviewer 1 Report
Minor form corrections are done in the text. I suggest an abreviation list to the end of the text because there are too much acronyms in the text.

Author Response
Point 1 – Minor form corrections are done in the text (L49; L525; L541). I suggest an abbreviation list to the end of the text because there are too much acronyms in the text.
Response 1 - We appreciated the Reviewer’s corrections and suggestions. We accordingly added an abbreviation list at the end of the text, also reducing the number of acronyms used.

Reviewer 2 Report
General comments
In this work, Dissegna and colleagues used three common techniques to identify the number of wolf packs residing on a protected area of the northern Apennines. They found that when sampling effort was high, all methods detected the same number of packs individually but when sampling effort was lower more techniques used in conjunction were required to identify a higher number of packs.
I enjoyed reading this manuscript and think it will be of interest to the broad readership of Genes. The authors should be commended on their huge sampling effort which is always challenging for wildlife. The paper is well written, concise, and informative with evidently sound methodology. I only had one major comment and that is how are you defining the best performing method? Throughout the manuscript it is discussed which method will perform the best but the measurement of success is not mentioned. As you do not know how many packs reside within the protected area, you cannot know what the truth is and therefore which method best resembles the truth. Is it possible to have false positives with pack detection and if so, is the highest number of packs detected really the most appropriate measure of method success? While I appreciate that knowing the truth to compare back to is not practical in wildlife studies, the authors need to introduce their metric for success of these methods and provide justification for why that is in the Introduction. This addition will greatly improve the paper. Some additional minor comments are provided below.
Minor comments
L103: Is it currently unknown how many packs occupy the park? Need to make this clear so people aren’t thinking you’ve got a baseline value to compare your methods back to.
L120: Define this acronym at the first mention in the Introduction.
L182-183: Are these microsatellite markers for differentiating between species also useful for differentiating between individuals? Update: I see this information is provided in the Supplementary Material. As your primary objective with this analysis is to identify individuals, this information needs to be moved to the main text.
Table 7: Could this table go in the Supplementary Material? The total cost is already provided in a previous table and you have quite a few display items already.
L406-407: No need to spell the acronyms out again, once they are defined, use them the same way throughout.
L407: How do you know that you are detecting the minimum number of packs? Is there a chance for false positives?
L495: Again, what defines the best performing result?
Author Response
Point 1 - In this work, Dissegna and colleagues used three common techniques to identify the number of wolf packs residing on a protected area of the northern Apennines. They found that when sampling effort was high, all methods detected the same number of packs individually but when sampling effort was lower more techniques used in conjunction were required to identify a higher number of packs.
I enjoyed reading this manuscript and think it will be of interest to the broad readership of Genes. The authors should be commended on their huge sampling effort which is always challenging for wildlife. The paper is well written, concise, and informative with evidently sound methodology.
Response 1 - Thank you very much for appreciating our work, the efforts to finalize it and its scientific interest.
Point 2 - I only had one major comment and that is how are you defining the best performing method? Throughout the manuscript it is discussed which method will perform the best but the measurement of success is not mentioned. As you do not know how many packs reside within the protected area, you cannot know what the truth is and therefore which method best resembles the truth. Is it possible to have false positives with pack detection and if so, is the highest number of packs detected really the most appropriate measure of method success? While I appreciate that knowing the truth to compare back to is not practical in wildlife studies, the authors need to introduce their metric for success of these methods and provide justification for why that is in the Introduction. This addition will greatly improve the paper. Some additional minor comments are provided below.
Response 2 - Thank you, we really appreciate this comment. We tried to integrate it in our study to improve its quality. Accordingly, we chose a more conservative title not mentioning a "best-performing method”, we changed some sentences throughout the manuscript paying particular attention to describe the merits and limits of each method in detecting wolf packs in the study area, and we better discussed the advantage of multiple approaches in studying population and pack dynamics in wolves.
Minor comments
Point 3 - L103: Is it currently unknown how many packs occupy the park? Need to make this clear so people aren’t thinking you’ve got a baseline value to compare your methods back to.
Response 3- Thank you, we have now specified that the number of packs was unknown during the sampling year of this study. The last official count of packs number was made by Caniglia et al. (2014), using non-invasive genetic sampling from 2002 to 2009. In that study they had counted 8 packs. We mentioned this information in the Survey design section of the manuscript.
Point 4 - L120: Define this acronym at the first mention in the Introduction.
Response 4 - Thank you, we added the acronym definition in the Introduction.
Point 5 - L182-183: Are these microsatellite markers for differentiating between species also useful for differentiating between individuals? Update: I see this information is provided in the Supplementary Material. As your primary objective with this analysis is to identify individuals, this information needs to be moved to the main text.
Response 5 - Thank you, we moved this information from Supplementary Material to the main text and we better clarified the utility of our STR panel in both individual identification and hybrid detection.
Point 6 - Table 7: Could this table go in the Supplementary Material? The total cost is already provided in a previous table and you have quite a few display items already.
Response 6 - Thank you, we moved this table in the Supplementary Material.
Point 7 - L406-407: No need to spell the acronyms out again, once they are defined, use them the same way throughout.
Response 7 - Thank you, we removed this second acronym.
Point 8 - L407: How do you know that you are detecting the minimum number of packs? Is there a chance for false positives?
Response 8 - Thank you for this question. For each method, we followed a highly conservative approach, thus the probability of false negatives is likely much higher than that of false positives. Moreover, the total concordance across methods in the IESA, the intensively monitored area, suggests that our count should be very close to the real number of living packs.
Point 9 - L495: Again, what defines the best performing result?
Response 9 - As we answered above, we changed some statements in order to clarify the role of each single method without assuming which was the best performing result.
